# [Re] CUDA: Curriculum of Data Augmentation for Long-tailed Recognition

**Barath Chandran.C**                                                                 *barath_cc@ece.iitr.ac.in*
*Indian Institute of Technology*
*Roorkee*

Reviewed on OpenReview: *https://openreview.net/forum?id=Wm6d44I8St*

## Abstract

In this reproducibility study, we present our results and experience while replicating the paper titled CUDA: Curriculum of Data Augmentation for Long-Tailed Recognition (Ahn et al., 2023). Traditional datasets used in image recognition, such as ImageNet, are often synthetically balanced, meaning each class has an equal number of samples. In practical scenarios, datasets frequently exhibit significant class imbalances, with certain classes having a disproportionately larger number of samples than others. This discrepancy poses a challenge for traditional image recognition models as they tend to favor classes with larger sample sizes, leading to poor performance in minority classes. CUDA proposes a class-wise data augmentation technique that can be used over any existing model to improve the accuracy for LTR: Long-Tailed Recognition. We successfully replicated a substantial portion of the results pertaining to the long-tailed CIFAR-100-LT dataset and extended our analysis to provide deeper insights into how CUDA efficiently tackles class imbalance.

## 1 Introduction

Long-tailed Recognition presents one of the most formidable challenges in visual recognition. This problem revolves around training highly effective models from datasets characterized by a large number of images distributed along a long-tailed distribution. In datasets characterized by LTD, a notable skew exists in the distribution of samples across different classes, resulting in certain classes being vastly over-represented (Head classes) while others are significantly under-represented (Tail classes). To elucidate this phenomenon, consider disease screening tests: the head class comprises of numerous samples of non-patients, whereas the tail class comprises of fewer samples of patients. In such scenarios, deep learning models often perform well on the head classes but struggle with the tail classes due to their limited representation.

Solutions to Long-tailed recognition primarily involve three methods: (1) Resampling (Buda et al., 2018): upsampling minority classes and downsampling majority classes. (2) Reweighting (Cao et al., 2019): rebalancing the loss to give more weight to minority classes. (3) Transfer learning (Kim et al., 2020): enriching the information of minority classes by transferring information gathered from majority classes to minority classes. While numerous strategies have been suggested to use data augmentation techniques for generating diverse representations of minority samples, minimal attention has been given to assessing the impact of varying augmentation degrees across different classes on addressing class imbalance issues.

The original authors proposed that applying an algorithm to determine class-wise augmentation strength can potentially address the imbalance problem in long-tailed visual recognition tasks. This data augmentation technique called CUDA, when used along with existing Long-Tailed Recognition (LTR) models, increases the performance of those models. The other key finding, as highlighted in the original paper, is that after training when we examine the class-wise augmentation strength, the majority classes have a stronger degree of augmentation and the minority classes have a weaker degree of augmentation. This finding is counter-intuitive, as one would typically anticipate that the minority class with fewer samples would undergo strong

augmentation, while the majority class with more samples would undergo weaker augmentation. The original authors utilized two metrics, weight L1-norm, and feature alignment gain, to demonstrate the effectiveness of CUDA in mitigating the imbalance problem. However, the original authors did not elaborate on how these two metrics are linked to the feature representation space of imbalanced datasets or elucidate how CUDA influences the feature representation space.

The motivation behind this reproducibility study is threefold: (1) To validate the assertion made by the original authors that employing class-wise augmentation strength can enhance the performance of existing LTR models. (2) To confirm the counter-intuitive observation from the original paper that employing stronger augmentation on majority classes and milder augmentation on minority classes yields superior model performance compared to the opposite strategy. (3) To delve deeper into how CUDA effectively addresses the imbalance problem by leveraging insights from prior research on feature representations in long-tailed datasets, a dimension not explored in the original paper.

## 2   Scope of Reproducibility

To address our threefold motivation behind this paper and to reproduce the results of the original paper, we perform the following experiments over the CIFAR-100-LT dataset:

1. For our first motivation, we examine the performance of CUDA across LTR models like CE (cross-entropy), CE-DRW (cross-entropy dynamic reweighting), LDAM-DRW (label-distribution-aware margin loss), BS (balanced softmax), and RIDE (Table-1).

2. For our second motivation, we investigate the LOL (Learning Objective Level) score, representing the augmentation strength of each class after training with CUDA (Figure-2).

3. We examine how accuracy changes with the three hyper-parameters augmentation probability, number of test samples, and acceptance rate to reproduce the result that both excessive and insufficient augmentation adversely affect performance (Figure-5).

4. We examine the metrics variance of weight L1-norm and feature alignment gain to reproduce the result that CUDA leads to a decrease in weight L1-norm and a positive feature alignment gain (Figure-3 and Figure-4).

5. We evaluate the contribution of curriculum learning and class-wise score on the performance of CUDA to reproduce the result that both curriculum learning and class-wise score are important to the performance of CUDA (Figure-6).

6. We compare the performance of CUDA with other augmentation methods to validate the result that CUDA outperforms all existing augmentation techniques (Figure-6).

7. We conduct the performance analysis across three different imbalance ratios (100, 50, 10) to examine how the performance of CUDA varies with dataset imbalance ratios (Table-2).

8. We compare how CUDA performs with and without cutout augmentation to quantify its contribution (Figure-8).

9. For our third motivation, we compare the feature representation space of the vanilla and CUDA versions, examining metrics such as inter-class distance and intra-class distance (Figure-7).

## 3   Methodology

### 3.1   Model descriptions

We use ResNet-32 as our backbone model for all 5 models: CE, CE-DRW, BS, LDAM-DRW, and RIDE. However, RIDE uses a modified version of ResNet-32 implemented according to the original paper. We

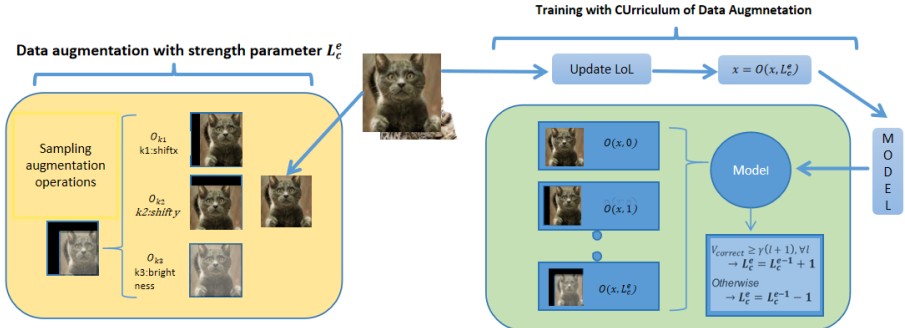

Figure 1: Schematic of CUDA. The left part depicts the strength-based augmentation and the right part depicts how the Level-of-Learning(LOL) score is updated in a given epoch

apply the data augmentation technique CUDA over these existing 5 models. The core philosophy of CUDA is to "generate an augmented sample that becomes the most difficult sample without losing its original information". CUDA utilizes two main parts to achieve this: (1) Strength-based augmentation and (2) Using Level-of-Learning (LOL) score. In the following sections 3.1.1 and 3.1.2, we explain the methodology of CUDA as outlined in the original paper.

### 3.1.1 Strength-based augmentation

We need to define a metric to quantify the degree of augmentation we are applying to an image. Let us assume that there exist $K$ predefined augmentation operations indexed as $k \in \{1, \ldots, K\}$, for example, gaussian blur, rotation, horizontal flip, etc. Each augmentation operation $\mathcal{O}_k^{m_k(s)} : \mathbb{R}^d \to \mathbb{R}^d$ has its own predefined augmentation magnitude function $m_k(s)$ with the strength parameter $s \in \{0, \ldots, S\}$. Given an augmentation strength parameter s and an input image x, we model a sequence of augmentation operations $O(x; s)$ as follows:

$$\mathcal{O}(x; s) = \mathcal{O}_{k_s}^{m_{k_s}(s)} \circ \mathcal{O}_{k_{s-1}}^{m_{k_{s-1}}(s)} \circ \cdots \circ \mathcal{O}_{k_1}^{m_{k_1}(s)}(x), \quad k_i \sim \mathrm{Cat}(K, \mathcal{U}(K)) \quad \forall i = \{1, \ldots, s\}$$

The sequential augmentation operation $\mathcal{O}(x; s)$ involves sampling s operations from a categorical distribution, where each operation is chosen from a uniform distribution among $K$ possible operations. In essence, out of the $K$ possible operations, only s operations are selected, each with a magnitude $m(3)$. For example, let's consider $s = 3$. Suppose the selected augmentation operations $k_1$, $k_2$, and $k_3$ correspond to brightness adjustment, x-shift, and y-shift, respectively. In this case, $\mathcal{O}(x; 3)$ outputs an image where the brightness is increased by $m_{bright}(3)$, shifted by $m_{x-shift}(3)$ on the x-axis, and shifted by $m_{y-shift}(3)$ on the y-axis. As $s$ increases, both the number of augmentation operations applied to each image and the magnitude of each of these operations increase consequently, the complexity of the augmentation process increases. Appendix A on augmentation preset describes how $m_k(s)$ the magnitude function of each data augmentation operation is defined.

### 3.1.2 Updating Level of Learning Score

To control the strength of augmentation properly, we need to check whether the model can correctly predict augmented versions without losing the original information. To enable this, we define the LOL for each class c at epoch e, i.e., $L_c^e$ which is adaptively updated as the training continues as follows:

- Initialization: At the start of training, set $L_c^e$ to zero for all classes. For clarity, let's consider that two epochs have already passed, and we begin at the third epoch. Let's assume $L_1^2$ the LOL score for a specific class 1 after the second iteration is 2.

- Update Mechanism: The LOL value for each class is updated using the function $V_{lol}$, which takes inputs such as the images $D_c$ belonging to class c, the previous LOL value $L_c^{e-1}$, the model $f_\theta$ used

for prediction, and hyperparameters $\gamma$ and T.

$$L_c^e = V_{\text{LoL}}\left(\mathcal{D}_c, L_c^{e-1}, f_\theta, \gamma, T\right)$$

- Sampling: Within $V_{lol}$ we iterate over all values of $l$ less than $L_1^2 = 2$. For each $l$ (0, 1, 2 in this case), randomly sample $T(l+1)$ samples from $D_c$ to form $D_c'$, ensuring $|D_c'| = T(l+1)$. Instead of utilizing all samples in set $D_c$ for prediction, we opt for a subset of images denoted as $D_c'$. The size of this subset is determined by the first hyperparameter, denoted as $T$, which represents the number of test samples.

- Calculation of $V_{correct}$: This involves summing the indicator function $1_{f_\theta(\mathcal{O}(x;l))=c}$ for all samples in $D'c$, where $\mathcal{O}(x;l)$ represents the application of augmentation strength $l$ to sample $x$. If the model correctly predicts class c (c=1 in this case), the function evaluates to 1. In essence, $V_{correct}$ quantifies the number of correct predictions among the $T(l+1)$ samples in $D_c'$.

$$V_{\text{Correct}}\left(\mathcal{D}_c, l, f_\theta, T\right) = \sum_{x \in \mathcal{D}_c'} \mathbb{1}_{\{f_\theta(\mathcal{O}(x;l))=c\}} \quad \text{where } \mathcal{D}_c' \subset \mathcal{D}_c$$

- Adjustment: If the number of correct predictions exceeds the threshold $\gamma \cdot T(l+1)$, where $l$ takes values from 0 to 2, it signifies that the ratio of correct predictions to total samples surpasses the second hyperparameter $\gamma$, representing the acceptance rate. In this scenario, the value of the LOL score for the next epoch, denoted as $L_1^3$, is incremented by 1 compared to the current value of $L_1^2$ ($L_1^3 = L_1^2 + 1 = 3$). If the condition is not met, the value of the LOL score is decremented by 1 ($L_1^3 = L_1^2 - 1 = 1$)

$$V_{\text{LoL}}\left(\mathcal{D}_c, L_c^{e-1}, f_\theta, \gamma, T\right) = \begin{cases} L_c^{e-1} + 1 & \text{if } V_{\text{Correct}}\left(\mathcal{D}_c, l, f_\theta, T\right) \geq \gamma T(l+1) \quad \forall l \in \left\{0, \ldots, L_c^{e-1}\right\} \\ L_c^{e-1} - 1 & \text{otherwise} \end{cases}$$

- Augmentation Sequence: By applying $V_{lol}$ for all classes we could then define a sequence of augmentation operations $O(x_i, L_{y_i}^e)$ with their strengths defined by the respective $L_{y_i}^e$ where each $y_i$ is a class.

- Augmentation Probability: Instead of completely replacing the training dataset $D$ with augmented images, we apply a sequence of augmentation operations with a probability specified by a third hyperparameter, $\rho$, representing the augmentation probability. This results in the creation of a new dataset $D_{CUDA}$.

- LTR algorithm: Execute any Learning to Rank (LTR) algorithm utilizing the dataset $D_{CUDA}$, comprising both augmented and original images.

## 3.2 Dataset and Hyper Parameters

The reproduction study is done on the dataset CIFAR-100-LT as mentioned in the previous works(Cao et al., 2019). The CIFAR-100 dataset (Canadian Institute for Advanced Research, 100 classes) is a subset of the Tiny Images dataset and consists of 60000 32x32 color images. Three different datasets are derived from the CIFAR-100 dataset with imbalance ratios 100, 50, and 10, where the imbalance ratio is defined as $|D_1|/|D_{100}| = N_{\max}/N_{\min}.|D_k|$ between $|D_1| = N_{\max}$ and $|D_{100}| = N_{\min}$ follows a linear decay. The hyperparameters—augmentation probability, number of test samples, and acceptance rate—are set according to the values stated in the original paper: 0.6 for augmentation probability, 10 for the number of test samples, and 0.5 for the acceptance rate. The hyperparameter sensitivity analysis done in Section 4.1.5 for the three hyperparameters further substantiated the values used.

### 3.3 Experimental setup and code

We have conducted all experiments for the CIFAR-100-LT dataset using the official repository, implemented in PyTorch. The code from the repository was reorganized into a Jupyter notebook to enhance portability and offer better control over the environment. Dependencies were not explicitly provided, and the versions of different libraries were determined through trial and error. Deprecated elements within the code were replaced with suitable, up-to-date alternatives. For training the model, parameters were set based on the specifications outlined in the paper. Any parameters not explicitly mentioned in the paper were assumed to use default values. The code for conducting component analysis on curriculum learning, class-wise score measurement, and measuring the standard deviation in weight L1 norm were re-implemented based on the specifications provided in the paper. The code and the readings are available here.

### 3.4 Computational requirements

We trained the model in Kaggle with 1 NVIDIA Tesla P100 as the GPU accelerator. The average training time of the model was approximately 40 minutes with a batch size of 128 for 200 epochs and the overall budget of the reproduction study was 250 GPU hours.

## 4 Results

### 4.1 Results reproducing original paper

Section 4.1.1 delves into the findings related to the first motivation of validating the assertion made by the original authors that employing class-wise augmentation strength can enhance the performance of existing LTR models. Section 4.1.2 delves into the findings related to the second motivation of confirming the counter-intuitive observation from the original paper that employing stronger augmentation on majority classes and milder augmentation on minority classes yields superior model performance compared to the opposite strategy. Section 4.1.3 and Section 4.1.4 provide additional insights into the original study on weight L1 norm and feature alignment gain that also justify our third motivation of investigating feature representations in long-tailed datasets. The subsequent sections in 4.1 encompass the results obtained from our reproduction of various analytical studies conducted in the original paper, aiming to further validate and understand the efficacy and implications of employing CUDA.

### 4.1.1 Comparison of validation accuracy

We measure the validation accuracy of CUDA when used with CE (Cross-Entropy), CE-DRW (Cross entropy Dynamic reweighting) (Cao et al., 2019), LDAM-DRW (label-distribution-aware margin loss) (Cao et al., 2019), BS (balanced softmax) (Ren et al., 2020), and RIDE (Wang et al., 2021) for the CIFAR-100-LT dataset following the general settings outlined in Cao et al. (2019). Specifically, we use ResNet-32 (He & Garcia, 2009) as the backbone network. The network is trained using stochastic gradient descent (SGD) with a momentum of 0.9 and a weight decay of 0.0002. The initial learning rate is set to 0.1, and a linear learning rate warm-up is applied during the first 5 epochs to reach the initial learning rate. The training process spans over 200 epochs, during which the learning rate is decayed at the 160th and 180th epochs by a factor of 0.01. The hyperparameters: acceptance rate, augmentation probability, and Number of test samples were 0.6, 0.5, and 10, respectively. The anticipated outcome was a consistent rise in validation accuracy across all five models and three imbalance ratios upon employing CUDA. In our reproduction of this study, we were able to reproduce the accuracy values with a maximum deviation of 2.4%. Further, we can observe from Table-1 a consistent increase in accuracy across all 5 models for the imbalance ratios 100 and 50 when paired with CUDA compared to the Vanilla edition. However, for the imbalance ratio 10, we see a minor improvement in accuracy for all models except when CUDA is paired with CE+CMO.

### 4.1.2 Dynamics of LoL Score

We plot the progression of LOL scores (the strength of augmentation) for various classes across five models: CE, CE-DRW, BS, LDAM-DRW, and RIDE. The anticipated outcome was a stark difference between the

| Algorithm | IR=100 | IR=50 | IR=10 | IR=100 | | |
|---|---|---|---|---|---|---|
| | | | | many | med | few |
| CE | 38.49 | 42.92 | 56.61 | 65.8 | 36.27 | 8.43 |
| CE+CMO | 42.08 | 46.93 | 60.08 | 68.87 | 41.23 | 11.7 |
| CE+CUDA | 42.27 | 46.91 | 59.79 | 70.47 | 41.93 | 9.6 |
| CE+CMO+CUDA | 43.01 | 48.14 | 58.62 | 70.2 | 42.47 | 11.77 |
| CE-DRW (Cao et al., 2019) | 40.86 | 46.22 | 57.83 | 62.33 | 41.2 | 15.23 |
| CE-DRW + Remix(Chou et al., 2020)† | 45.8 | 49.5 | 59.2 | ∼ | ∼ | ∼ |
| CE-DRW + CUDA | 47.52 | 50.57 | 58.18 | 66.43 | 50.27 | 22.03 |
| LDAM-DRW(Cao et al., 2019) | 42.48 | 47.15 | 57.96 | 62.17 | 42.43 | 19.4 |
| LDAM + M2m(Kim et al., 2020)‡ | 43.5 | ∼ | 57.6 | ∼ | ∼ | ∼ |
| LDAM+CUDA | 47.52 | 50.57 | 58.18 | 66.43 | 50.27 | 22.03 |
| BS | 42.24 | 45.99 | 58.29 | 59.63 | 41.33 | 22.8 |
| BS+CUDA | 47.47 | 52 | 61.52 | 63.03 | 48.8 | 27.83 |
| RIDE(Wang et al., 2021)† | 48.6 | 51.4 | 59.8 | ∼ | ∼ | ∼ |
| RIDE | 48.87 | 51.95 | 59.13 | 67.7 | 50.17 | 25.27 |
| RIDE+CMO(Park et al., 2022)† | ∼ | 53 | 60.2 | ∼ | ∼ | ∼ |
| RIDE+CMO | 49.61 | 52.78 | 59.91 | 68.03 | 50.9 | 26.47 |
| RIDE+CUDA | 49.53 | 53.58 | 61.21 | 68.23 | 50.7 | 24.63 |

Table 1: Validation accuracy on CIFAR-100-LT dataset. † are from Park et al. (2022) and ‡,* are from the original papers (Kim et al. (2020); Zhu et al. (2022)).We report the average results of three random trials on each scenario. The rows highlighted in green refer to the versions where CUDA is used. We can observe an increase in accuracy when we compare similar models with and without CUDA. For example, vanilla CE+CMO for imbalance ratio 100 has an accuracy of 42.08 whereas CE+CMO+CUDA has an accuracy of 43.01.

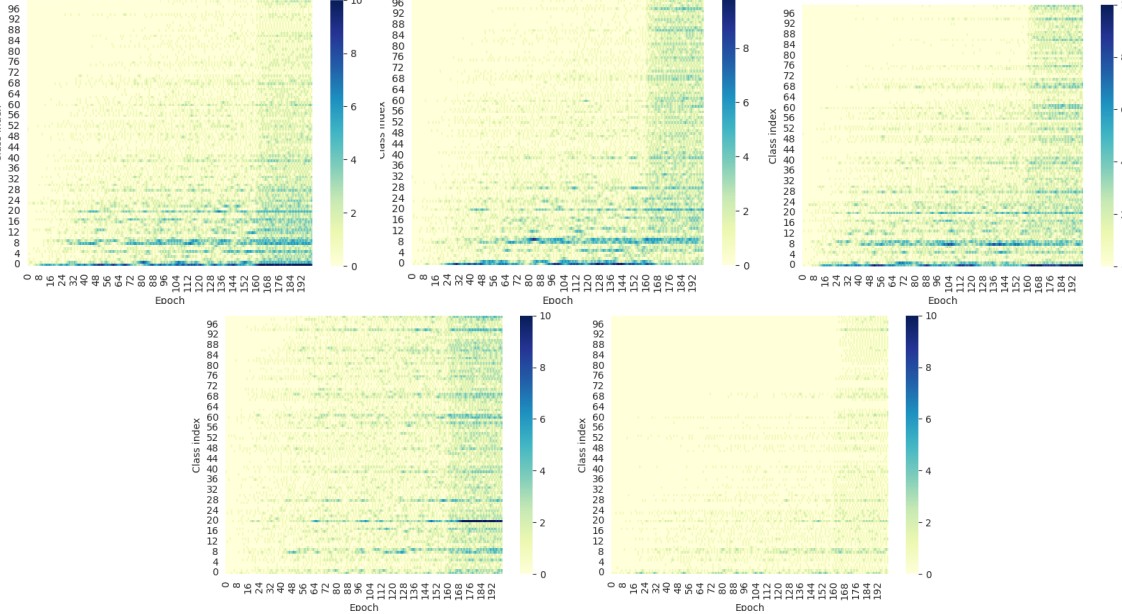

Figure 2: Evolution of LOL score in the order CE, CE-DRW, LDAM-DRW, BS, and RIDE over the epochs. The x-axis represents the epochs, with the 200th epoch positioned on the rightmost side of each graph. The y-axis will display the classes, arranged in descending order based on the number of samples in each class. The intensity of the color of the heatmap represents the strength of augmentation. It is evident that during training the majority classes(0-49) have a stronger augmentation compared to the minority classes(50-99).

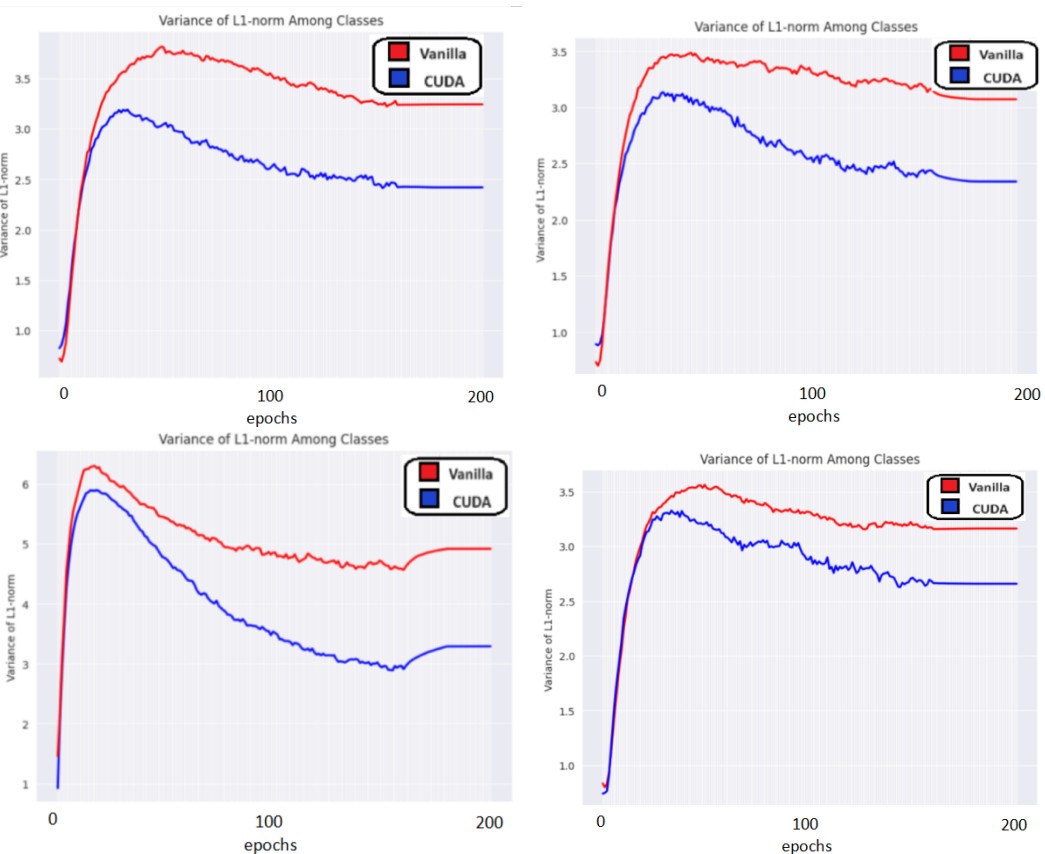

Figure 3: Variation Of weight-L1 norm in the order CE, CE-DRW, LDAM-DRW, BS over the epochs. The value of variance at epoch 200 represents the final value of variance of weight-L1 norm after training CUDA (indicated by the blue line) exhibits a lower variance of weight-L1 Norm compared to the vanilla version (depicted by the red line).

LOL scores of the majority class (0-49) and the minority class (50-99) with the LOL score of the majority class being greater. In our replication of this study, we observed that there isn't a steady decrease in augmentation strength as we move along the y-axis. However, there is a noticeable pattern of higher average augmentation strength for the majority classes (0-49) in comparison to the minority classes (50-99). The heat maps presented in Figure-2 validate the assertion that "Stronger augmentation on majority classes and weaker augmentation on minority classes yield better performance".

### 4.1.3 Variance of Weight L1-norm

Image recognition can be considered as a coupling of two tasks: feature learning where we extract features from the images and embed them into a feature space, and classifier learning where we train a classifier over the learned features (Kang et al., 2020). Long-tailed data distribution tends to have a skewed representation space, where the distance between head and tail categories is much larger than the distance between two tail categories (Fu et al., 2023). A naively trained model on long-tailed distributed data tends to have "artificially" large weights for the head classes. This yields a wider classification boundary in feature space for the head classes, allowing the classifier to have much higher accuracy on head classes, but hurting the performance of the tail classes (Kang et al., 2020).

The variance of the classifier weight norm is usually used to measure how balanced the classifier is considering the input from a class-wise perspective (Kang et al., 2021). A lower variance in weight L1 norm indicates that the classifier assigns similar importance to all classes. We analyze how CUDA affects the standard deviation

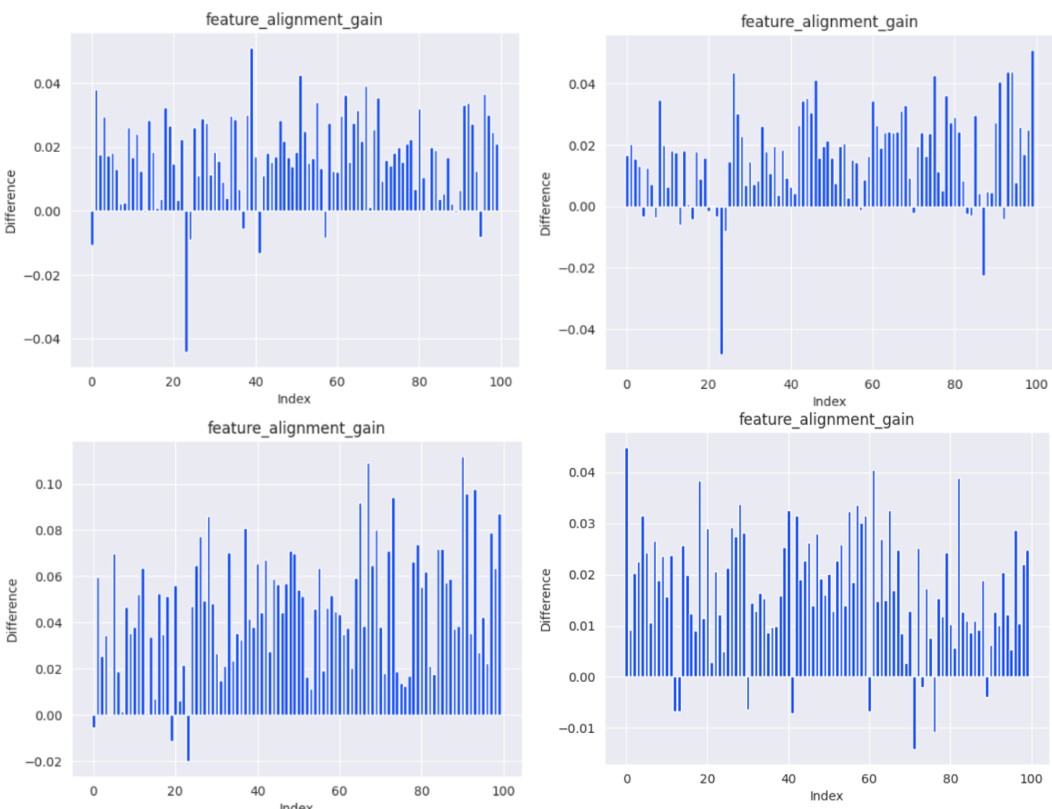

Figure 4: Feature alignment gain in the order CE, CE-DRW, LDAM-DRW, BS. Feature alignment gain is the increase in the average feature cosine similarity between feature vectors belonging to the same class. The x-axis represents the classes and the y-axis represents the feature alignment gain for that class. CUDA exhibits a positive feature alignment gain(indicated by the blue line) for most of the classes across all 4 models.

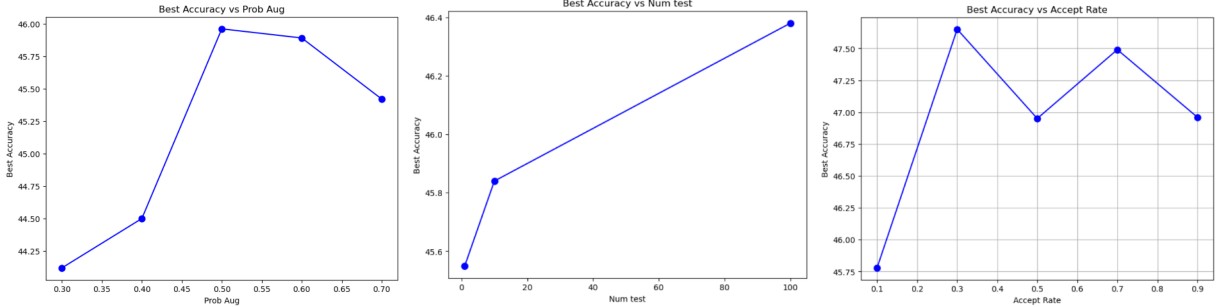

Figure 5: Hyperparameter Analysis of CUDA when paired with the model LDAM-DRW. The leftmost graph represents the sensitivity of accuracy with Augmentation probability. We see a concavity in performance when the value of Prob Aug is changed between 0.3 to 0.7. The middle graph represents the sensitivity of accuracy with the Number of test samples. We see a slight change in accuracy when the value of the Num Test is changed. The rightmost graph represents the sensitivity of accuracy with the Acceptance rate. We see a concavity in performance when the Acceptance rate value is changed between 0.1 to 0.9.

of weight L1 norm across the 4 models over the epochs. The anticipated result was a consistent reduction in the value of the variance of weight L1 norm on using CUDA across all 4 models. In our reproduction of this study, we observe a significant decrease in the standard deviation of weight L1 norm when we use CUDA compared to the vanilla version from Figure-3, validating the results of the original study.

We can say any enhancements in performance can be attributed to either improved feature representation or a more effective classifier. In this case, since the classifier remains the same for both vanilla and CUDA versions, the improvements are likely due to CUDA optimizing the feature representation space. The variance in weight norm decreases, implying a reduction in the distinction between different classification boundaries (between two head classes, or between two tail classes, or between a head and a tail class). Specifically, we can infer that CUDA is able to make the feature representation more balanced in terms of inter-class distances (Song et al., 2015).

### 4.1.4 Feature Alignment Gain

Continuing from the discussion of an imbalanced feature space representation in subsection 4.1.3, another significant impact of class imbalance on the feature representation space is the vulnerability of a cluster containing instances of a tail class (Huang et al., 2016). Due to their sparse nature these clusters, which are expected to contain instances of the same class, are more prone to invasion by imposter feature vectors from other classes present in the neighborhood.

Feature alignment gain serves as a metric to quantify the effectiveness of a feature extractor in embedding features into the feature space. Specifically, it measures the improvement in feature alignment, which is the sum of cosine similarities between pairs of feature vectors within a given class when CUDA is used. We analyze how CUDA affects the feature alignment gain of each class across the 4 models for the validation dataset. The anticipated result was mostly positive feature alignment gain i.e. a gain in feature alignment on using CUDA across all 4 models. In our reproduction of this study, we observe from Figure-4 that feature alignment gain is positive for most of the classes when we use CUDA.

When feature vectors within a class exhibit high similarity, they tend to cluster closely together within the feature representation space. This closeness results in reduced intra-class distance (Song et al., 2015), meaning that feature vectors belonging to the same class are positioned closer to each other. Consequently, the likelihood of intrusions from feature vectors of a foreign class is diminished.

### 4.1.5 Hyper-Parameter Analysis

The paper proposed training on RIDE, but we opted for LDAM-DRW as LDAM-DRW showed a higher increase in performance when paired with CUDA, which means that the magnitude of change in performance with a change in hyperparameter will be higher. For the acceptance rate, we trained the model LDAM-DRW with CUDA for 5 equally spaced values from 0.1 to 0.9. An acceptance rate of 0.1 means that the threshold for accepting the augmented samples is low, leading to higher augmentation strength. An acceptance rate of 0.9 means that the threshold for accepting the augmented samples is high, leading to lower augmentation strength. For augmentation probability, we trained the model LDAM-DRW with CUDA for 5 equally spaced values from 0.3 to 0.7. An augmentation probability of 0.3 means that most of the original images are retained when forming the data loader. An augmentation probability of 0.7 means that most of the original images are replaced by the augmented images when forming the data loader. The anticipated result was a concavity in the value of validation accuracy when the values of acceptance rate and probability of augmentation are increased. In our reproduction of the hyperparameter analysis studies from Figure-5, we can observe that acceptance rate and probability augmentation show a concavity in the performance. For the hyperparameter number of test samples, we trained the model LDAM-DRW with CUDA for three different values of 1, 10, and 100. The anticipated result was a steep increase in accuracy between 1 to 10 values of T and a slight increase between 10 to 100 values of T. In our reproduction of the hyperparameter analysis, we can see that the performance increases slightly with an increase in the number of test samples from Figure-5.

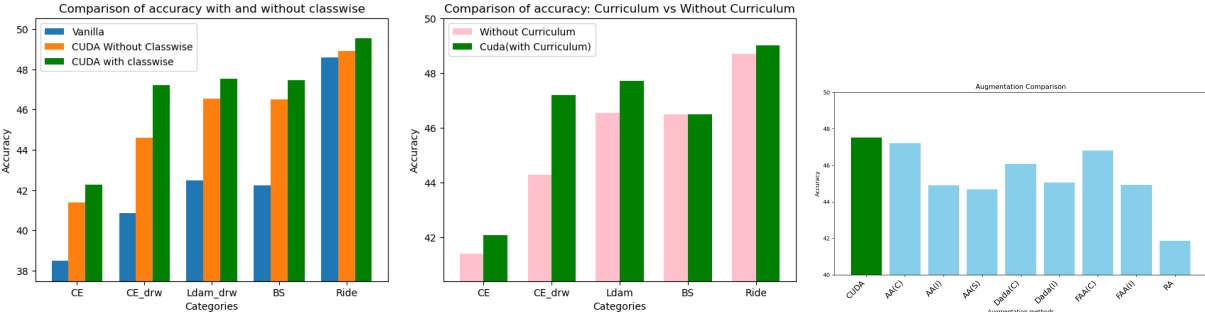

Figure 6: Component analysis on Class-wise Score and Curriculum Learning, Comparison of accuracy on different augmentation techniques. The leftmost graph illustrates the comparison of accuracy among Vanilla, CUDA without a Class-wise Score, and the original CUDA version with a Class-wise Score. The middle graph demonstrates the comparison of accuracy between CUDA without Curriculum Learning and the original CUDA version with Curriculum Learning. The rightmost graph represents the comparison of accuracy for 9 different augmentation methods: CUDA, Auto Augmentation-CIFAR policy, Auto Augmentation-ImageNet policy, Auto Augmentation-SVHN policy, Dada-CIFAR, Dada-ImageNet, Fast Auto Augmentation-CIFAR, Fast Auto Augmentation-ImageNet policy, and Random Augmentation. We observe that the original iteration of CUDA (represented by the green bar) consistently outperforms its competitors across all three studies.

### 4.1.6 Curriculum Learning

Curriculum Learning is a training strategy designed to enhance machine learning models by progressively exposing them to increasingly complex or challenging data during training. Previous works (Zhou et al., 2020) have shown that curriculum learning can improve the accuracy of LTR models. In the context of the CUDA algorithm, the LOL (Learning Objective Level) scores for each class initially start at zero and are iteratively updated at the end of each epoch based on the model's performance with augmented images. After 200 epochs, an optimal combination of LOL scores is achieved, leading to the final model performance. To assess the impact of curriculum learning on accuracy, a two-step approach is employed. In the first step, a model is trained using the standard CUDA procedure. Subsequently, the LOL scores obtained from this initial training run are extracted and utilized as fixed scores in a subsequent run. In this second run, the model is trained without updating the LOL scores. The anticipated result was a reduction in accuracy on the second run when CUDA is trained without curriculum learning. In our reproduction of this component analysis, we can observe from Figure-6 that there is a decrease in performance across all 5 models when CUDA is trained without curriculum learning.

### 4.1.7 Classwise Score

To examine the validity of class-wise augmentation of CUDA, we apply CUDA with the same strength of DA for all classes. Instead of computing the LOL score class-wisely, we computed only one LOL score for the entire dataset by uniformly random sampling instances in the training dataset regardless of class. The anticipated result was a reduction in accuracy when CUDA is trained without the class-wise score. In our reproduction of this component analysis, we can observe from Figure-6 a significant performance degradation of CUDA across all 5 models without class-wise score compared to CUDA.

### 4.1.8 Comparison with other Augmentation techniques

We compare the performance of CUDA with other augmentation techniques: Auto-Augmentation (CIFAR, ImageNet, and SVHN policy) (Cubuk et al., 2019), Fast Auto Augmentation (CIFAR, ImageNet, and SVHN policy) (Lim et al., 2019), DADA (Li et al., 2020), and Rand-Augmentation (m=1, n=2) (Cubuk et al., 2020) for the model LDAM-DRW. The anticipated result was that CUDA outperforms all other existing data

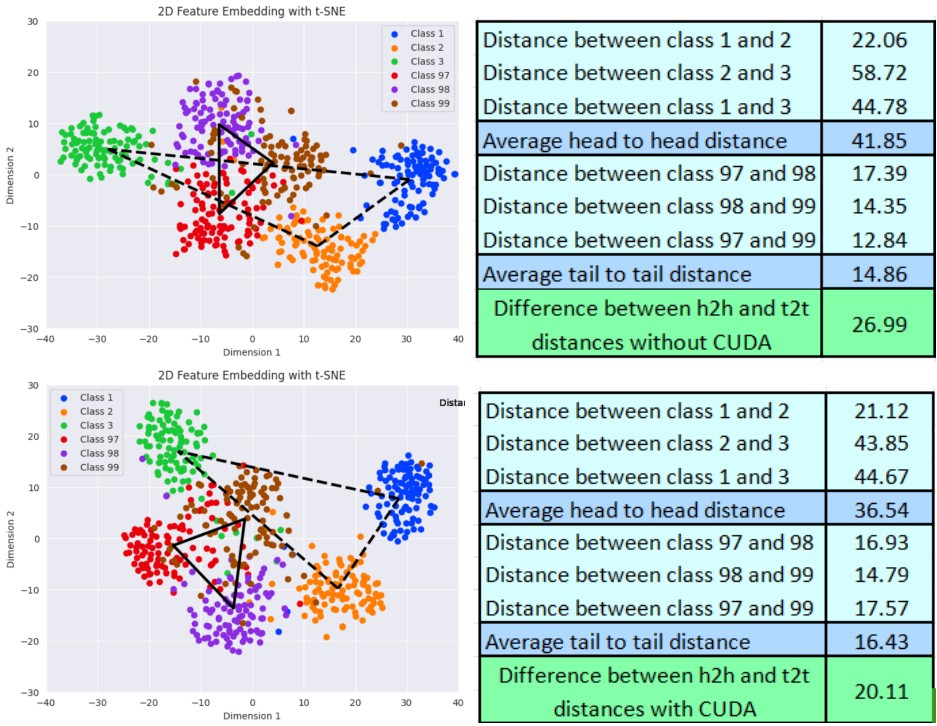

Figure 7: Leftmost - Feature representation of vanilla CE, inter-class distances for vanilla CE, Feature representation of CE+CUDA, rightmost - inter-class distances for CE+CUDA. In the feature representation space, the red, purple, and brown instances represent three tail classes (97, 98, 99) and the green, yellow, and blue instances represent three head classes (1, 2, 3). The dotted line connects the means of 2 head classes, and the solid line connects the means of 2 tail classes. The distances in the table are measured between the means of the classes. It is evident that after using CUDA, the difference between average head-to-head distance (h2h) and average tail-to-tail distance (t2t) has diminished.

augmentation techniques. In our reproduction of this comparison, Figure-6 reveals that CUDA consistently outperforms all other augmentation techniques.

## 4.2 Results beyond the original paper

Section 4.2.1 delves into findings related to our third motivation of exploring the changes in feature representation space after using CUDA. Section 4.2.2 consists of an analysis of the performance of CUDA for the augmentation operation cutout which was implemented in the original code but not explicitly stated to be used in the paper. Section 4.2.3 consists of an analysis of accuracy gain when CUDA is used across three different imbalance ratios.

### 4.2.1 Quantitative and Qualitative analysis of feature embedding

We analyzed the feature embedding derived from the validation dataset of the imbalanced CIFAR-100-LT dataset, building upon prior methodologies (Huang et al., 2016; Song et al., 2015) aimed at addressing class imbalance through enhanced feature representation. Specifically, we visualized the 2-dimensional feature representations of three head classes (1, 2, 3) and three tail classes (97, 98, 99) using t-SNE.

The quantitative aspect we examined was the variance in inter-class distances, as detailed in section 4.1.3. This variance can be quantified by the difference between the average distance within head classes and that within tail classes. Notably, we observed a diminished difference from the tables in Figure-7 when employing CUDA compared to the vanilla version, indicating an improvement in handling class imbalance. We also

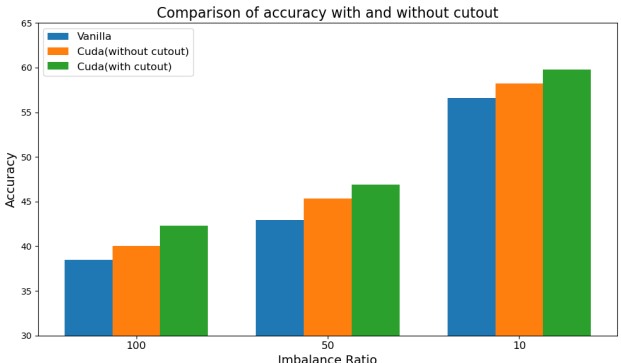

Figure 8: The image represents the comparison of the accuracy of CUDA when paired with cutout, when not paired with cutout, and the vanilla version.

| Algorithm | IR=100 | IR=50 | IR=10 |
|---|---|---|---|
| CE+CMO+CUDA | 0.93 | 1.23 | -1.46 |
| CE-DRW+CUDA | 6.37 | 5.68 | 3.93 |
| LDAM-DRW+CUDA | 5.04 | 3.42 | 0.22 |
| BS+CUDA | 5.23 | 6.01 | 3.23 |
| RIDE(3 experts)+CUDA | 0.66 | 1.63 | 2.08 |

Table 2: The table represents the analysis of gain in accuracy with 3 different versions of CIFAR-100-LT with an imbalance ratio:10,50, and 100.

observe a slight increase in the average distance of means of tail classes denoting a wider classification boundary than before between tail classes.

The qualitative aspect we examined was the representation of the tail class clusters in the feature space. Following the adoption of CUDA, we noted that the clusters corresponding to tail classes exhibited greater coherence from Figure-7. There was a reduction in the intrusion of imposter vectors into these clusters, signifying enhanced separability.

### 4.2.2 Cutout

The usage of the augmentation operation Cutout was not explicitly mentioned in the original paper. However, the official implementation of CUDA also had an argument to either use or not use the cutout operation. Cutout has proved to improve the generalization performance of CNNs (DeVries & Taylor, 2017). Our analysis focused on investigating how Cutout affects the performance of CUDA for the CE (cross-entropy) model to validate the fact that the original authors used the cutout augmentation operation during training for all models. Our findings in Figure-8 suggest that incorporating the cutout augmentation operation during training is crucial for achieving the reported accuracy levels stated in the original paper for all models.

### 4.2.3 Gain in accuracy for imbalance ratio

We conducted a comparative analysis of the gain in accuracy among five models when paired with CUDA across three different imbalance ratios: 100, 50, and 10. Imbalance ratios reflect the disparity in class distribution within the dataset, with higher ratios indicating more pronounced class imbalances. Our findings in Table-2 reveal a general trend where the gain in accuracy diminishes as the imbalance ratio decreases. We can infer that CUDA performs better when the imbalance ratio of the dataset is higher.

# 5 Discussion

The experimental results reported in this study effectively support the first claim of the original paper, indicating that using an algorithm to find class-wise augmentation strength can enhance validation accuracy (section 4.1.1). However, due to time constraints, the performance of CUDA with model BCL could not be studied. Despite this limitation, we can still validate the first claim, which states that employing a class-wise degree of augmentation improves performance based on available data. The heatmaps in Figure-2 represent the growth of augmentation strength over the epochs, serving to validate the original paper's second assumption. Before the 160th epoch, there was a stark difference in LOL scores between the majority classes (0-49) and minority classes (50-100). However, after 200 epochs of training, this contrast is greatly reduced. A holistic view of the evolution of LOL scores across epochs (section 4.1.2) still confirms the original paper's second claim, that using stronger augmentation on majority classes and milder augmentation on minority classes results in better model performance than the opposite strategy.

By measuring the variation of weight L1-norm (section 4.1.3), the reproduction study demonstrates how CUDA effectively addresses the imbalance problem by reducing the variance of weight L1-norm, helping the model to give equal importance to each class as claimed in the original paper. Through further analysis, we inferred that CUDA improves the balance of feature representation in terms of inter-class distances. In addition, we replicated the feature alignment gain analysis (section 4.1.4), confirming CUDA's effectiveness in mitigating the imbalance issue by boosting feature alignment, as stated in the original paper. Through further analysis, we inferred that CUDA minimizes intra-class distances between instances of the same class, particularly the tail classes. To validate these two inferences regarding balanced inter-class distances and lowered intra-class distances, we compared the feature representation space of the vanilla and CUDA versions (section 4.2.1). Our findings supported both hypotheses, demonstrating that CUDA improves feature representation by promoting balance in inter-class distances and decreasing the invasion of foreign instances by reducing intra-class distances.

The hyperparameter sensitivity analysis for augmentation probability and acceptance rate on validation accuracy demonstrates the concavity claimed in the original paper (Figure-5). Our findings in (sections 4.1.6 and 4.1.7) demonstrate the impact of curriculum and class-wise score on CUDA performance, verifying their importance as claimed in the original paper. We discovered that cutout augmentation plays a significant role in the accuracy gain observed when using CUDA, even more so than curriculum learning or class-wise score (section 4.2.3). We claim the original authors had used cutout augmentation during training across all models but have not explicitly stated this in the paper. The stated accuracy values in the original paper can only be attained when CUDA is combined with cutout augmentation. This dependency may limit the generalizability of CUDA across different scenarios. Our analysis of the imbalance ratio indicates that CUDA's efficiency diminishes as we move towards datasets with lower imbalance (section 4.2.3). From this, we can also infer that CUDA is not beneficial when dealing with balanced datasets. The reliance on a larger imbalance ratio and cutout augmentation is a shortcoming in the CUDA methodology.

## 5.1 What was easy and what was difficult

The original paper was really easy to follow. The section on the repository for CIFAR-100-LT was written clearly. The description of the arguments that we pass during training was properly stated. The algorithm that makes up CUDA was completely logical in its implementation. The lack of significant barriers in setting up the code enhances its portability. The dependencies were not mentioned clearly by the author requiring additional time to find the versions by trial and error. The original paper also included a performance comparison on datasets ImageNet-LT and Inaturalist-18, the section of the original repository for ImageNet-LT and Inat-18 contains redundant code and uncleaned up code. The paper claims to deviate from the model recipes of BCL and NCL to ensure a fair comparison. However, it fails to clearly state these deviations, making it difficult to assess their performance in this study.

## 5.2 Communication with original authors

The initial attempts to contact the authors through the email IDs given in the paper were not successful. We were able to contact the authors through Linked In in the latter half of the study. The authors were able to clarify our doubts about implementing the component analysis for curriculum learning and class-wise scores. Regrettably, the authors were unable to provide a clear recipe for BCL.

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

## A   Augmentation preset

We use 22 different augmentation operations for CUDA each having its parameter. The details of each of these operations have been described in Table 1. The magnitude parameter divides the augmentation parameter into 30 values linearly. For example, if the max value for Rotate is 30 and the min value is 0, the magnitude of the parameter for Rotate is defined by

$$m_{\text{rotate}}(s) = (30 - 0)/30 * s, \text{ thus } m_{\text{rotate}}(1) = 1 = (30 - 0)/30 * 1$$

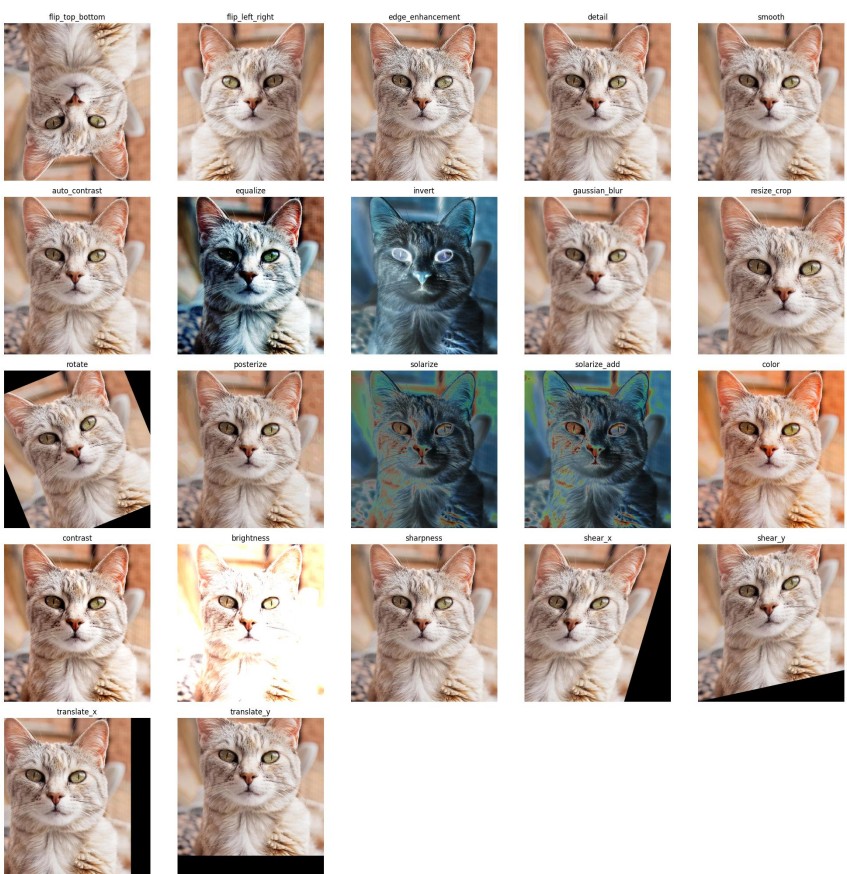

Figure 9: The 22 augmentation operations we use for CUDA

Table 3: Operation Parameter Description

| Operation | Description |
|---|---|
| Flip On/Off | Flip top and bottom |
| Mirror On/Off | Flip left and right |
| Edge Enhancement On/Off | Increasing the contrast of the pixels around the targeted edges |
| Detail On/Off | Utilize convolutional kernel $\begin{bmatrix} 0 & -1 & 0 \\ -1 & 10 & -1 \\ 0 & -1 & 0 \end{bmatrix}$ |
| Smooth-On/Off | Utilize convolutional kernel $\begin{bmatrix} 1 & 1 & 1 \\ 1 & 5 & 1 \\ 1 & 1 & 1 \end{bmatrix}$ |
| AutoContrast On/Off | Remove a specific percent of the lightest and darkest pixels |
| Equalize On/Off | Apply non-linear mapping to make uniform distribution |
| Invert On/Off | Negate the image |
| Gaussian Blur | Blurring an image using Gaussian function with radius [0,2] |
| Resize Crop | Resizing and center random cropping with scale [1,1.3] |
| Rotate | Rotate the image with angle [0,30] |
| Posterize | Reduce the number of bits for each channel in the range [0,4] |
| Solarize | Invert all pixel values above a threshold in the range [0,256] |
| SolarizeAdd | Adding value and run solarize in the range [0,110] |
| Color | Colorize grayscale values in the range [0.1, 1.9] |
| Contrast | Adjust the distance between colors in the range [0.1,1.9] |
| Brightness | Adjust image brightness in the range [0.1,1.9] |
| Sharpness | Adjust image sharpness in the range [0.1,1.9] |
| Shear X | Shearing X-axis in the range [0,0.3] |
| Shear Y | Shearing Y-axis in the range [0,0.3] |
| Translate X | Shift X-axis in the range [0,100] |
| Translate Y | Shift Y-axis in the range [0,100] |

