# OpenReview forum: "[Re] CUDA: Curriculum of Data Augmentation for Long‐tailed Recognition"
_TMLR — Accepted by TMLR_

### Review · Reviewer_MBg6 · 2024-03-10

**Summary Of Contributions:**

This paper replicated the methodology outlined in the CUDA paper; Conducted experiments on the Cifar100-LT dataset to evaluate the effectiveness of the proposed CUDA technique; and analyzed and compared the performance of various models with and without CUDA augmentation.

**Audience:**

No

**Claims And Evidence:**

Yes

**Requested Changes:**

1. The absence of tags on the red and blue curves in Figure 5 creates ambiguity regarding their meanings.
2. Figure 3 illustrates that adding CUDA to the method CE +CMO did not result in improvement for the IR = 10. Why claim observe a consistent increase in accuracy?

**Strengths And Weaknesses:**

Strength:
1. The method description and experimental setup are meticulously clear, offering an understanding of the CUDA technique and its implementation.

Weaknesses:

1. Lack of insight is evident in certain sections, such as Section 4.1.3, where the paper claims a significant decrease in the standard deviation of weight L1 norm. It is essential to delve deeper into these observations to provide meaningful insights into the underlying mechanisms or implications of such findings.

2. There is room for additional evaluation or experimentation, particularly regarding the study on class-wise feature cosine similarity. Exploring this aspect further could potentially yield valuable insights and discoveries that could benefit the broader research community.

3. The motivation behind reproducing the original paper is not adequately clarified. It's essential to elucidate the significance and relevance of replicating the study to the audience, outlining the key contributions or insights that the replication study aims to offer.

---

> ### Author Response · Authors · 2024-04-05
> **Response to reviewer MBg6 part1/3**
>
> #### We are very grateful for your constructive comments. We express your concerns as we understood and answer them as follows. We were able to restructure our paper, conduct further experiments and add more insights based on you review, with the intention of making the paper more beneficial to the audience. We answer the questions by reordering them in the order in which we thought they were highly correlated to help your understanding and based on the order in which we made the changes. Please consider re-evaluating the paper if you are satisfied with the rebuttal. Please let us know through further comments if there are any misunderstandings or if you have any additional questions or any additional changes to be made in the paper. Links to relevant papers have been formatted in the comment.
>  ___
>
>  #### **1)Concern regarding lack of evaluation of feature cosine similarity**
> #### We were able to conduct the analysis on feature alignment gain as stated in the original paper. The results can be seen in section 4.1.4. We analyzed how CUDA affects feature alignment gain of each class across the 4 models for the validation dataset. The anticipated result was a mostly positive feature alignment gain on using CUDA across all 4 models. In our replication we got a positive feature alignment gain for most of the classes when we use CUDA. After understanding that feature alignment gain is effectively just a measure of how close the instances belonging to the same class are, we came to know about a metric called intra class distance from a previous paper on LTR [*song2015deep*](https://arxiv.org/abs/1511.06452).Intra class distance measures the distance between instance belonging to the same class in feature representation space. We made a deduction that CUDA is able to decrease intra class distance which was our first deduction.
>  ___
>
> #### **2)Concern regarding Lack of insight in Section 4.1.3 (Variance of Weight L1 norm)**
> #### The paper [*song2015deep*](https://arxiv.org/abs/1511.06452) also contained another metric called inter class distance, the distance between means of two classes in the feature representation space. [*Kang2020Decoupling*](https://arxiv.org/abs/1910.09217) says that having a wider classification boundary between head classes in the feature representation space of an imbalanced dataset is the reason that head classes have better accuracy than tail classes.[*kang2021exploring*](https://openreview.net/forum?id=OqtLIabPTit) states that the reason we measure variance of weight L1 norm is that if our weight norms of each class are similar then our classification boundaries in the feature representation space is also similar. Logically we can say that inter class distance is a measure of the width of our classification boundary. Since CUDA is able to reduce variance in weight L1 norm, we made a deduction that CUDA must make the feature representation space relatively more balanced than vanilla version in terms of inter class distances and this was our second deduction.
>  ___
>
> #### **3)Combining concerns 1 and 2**
> #### [*song2015deep*](https://arxiv.org/abs/1511.06452) states that decreasing intra class distance and improving inter class distance will improve the performance of recognition models on long tailed data. To validate both of the deduction we decided to analyze the feature representation space of CE before and after using CUDA. We used t-SNE to get our feature representation space as stated in the paper  [*huanglearning*](https://ieeexplore.ieee.org/document/7780949).We did a qualitative analysis for intra class distance by just comparing the amount of invasion of foreign instances in the cluster belong to a specific tail class(let say red). We observed after using CUDA the amount of invasions decreased. We did a quantitative analysis for inter class distance by comparing the average distance between 3 head and 3 tail classes. We observed a decrease in the difference between average h2h and t2t distance.h2h=head to head , t2t=tail to tail. Hence we were able to validate both of our deductions. We restructured our paper to include this analysis of feature space(which was not part of the original paper) to be our third motivation.

---

> ### Author Response · Authors · 2024-04-05
> **Response to reviewer MBg6 part 2/3**
>
> #### Kindly read part 1/3 before to improve understanding.
>
> #### **4)Concern regarding lack of clarity in motivation**
> #### We made three stages of changes in the paper to clarify the motivation behind the paper. The text in italic are taken directly from paper.
>
> - #### **Introduction**
> #### In  introduction, we emphasized why the two key findings of the original paper are so crucial. The two findings are that just applying a classwise degree of augmentation can enhance LTR, and that applying stronger augmentation to majority classes and weaker augmentation to minority classes is superior to the opposite approach. Our motive is to validate these two facts (one for each).Our third motivation is to investigate the feature representation space for the reasons described in Part 1/3.We clearly declared our motivation after properly presenting the reasons for it.
>
> #### *The original authors propose that applying an algorithm to determine class-wise augmentation strength can potentially address the imbalance problem in long-tailed visual recognition tasks. This data augmentation technique called CUDA is designed to complement existing Long-Tailed Recognition (LTR) models. The other key finding, as highlighted in the original paper, is that after training when we examine the class-wise strength of augmentation the majority classes have a stronger degree of augmentation and the minority classes have a weaker degree of augmentation. This finding is counter-intuitive, as one would typically anticipate that the minority class, with fewer samples, would undergo strong augmentation, while the majority class, with more samples, would undergo weaker augmentation. The original authors utilized two metrics, weight L1-norm and feature alignment gain, to demonstrate the effectiveness of CUDA in mitigating the imbalance problem. However, the original authors did not elaborate on how these two metrics are linked to the feature representation of imbalanced datasets or elucidate how CUDA influences the feature representation space.*
>
> #### *The motivation behind this reproducibility study is threefold: (1) To validate the assertion made by the original authors that employing class-wise augmentation strength can enhance performance of existing LTR models. (2) To confirm the counter-intuitive observation from the original paper that employing stronger augmentation on majority classes and milder augmentation on minority classes yields superior model performance compared to the opposite strategy. (3) To delve deeper into how CUDA effectively addresses the imbalance problem by leveraging insights from prior research on feature representations in long-tailed datasets, a dimension not explored in the original paper.*
>
> - #### **Scope of Reproducibility**
> #### We have updated the scope of reproducibility section to correlate the experiments with the corresponding motivations.
>
> #### 1. *For our first motivation, we examine the performance of CUDA across LTR models like CE (Cross Entropy), CE-DRW (Cross entropy Dynamic reweighting) , LDAM-DRW (label-distribution-aware margin loss) , BS (balanced softmax) and RIDE(Figure 3).*
> #### 2. *For our second motivation, we investigate the LOL (Learning Objective Level) score, representing the augmentation strength of each class after training with CUDA(Figure 4).*
> #### 8. *For our third motivation, we compare the feature representation space of the vanilla and CUDA versions, examining metrics such as inter-class distance and intra-class distance(Figure 9).*
>
> - #### **Discussion**
>
> #### We have modified the discussion section to clarify whether the 3 motivations behind the paper were achieved or not. We added statements such as
> #### *" The experimental results presented in the paper effectively supports the first claim of the original paper, demonstrating that applying an algorithm to find classwise augmentation strength can show improvements in validation accuracy (section 4.1.1)"*
> #### *"A holistic view of the evolution of LOL scores across epochs (section 4.1.2) still corroborates the second claim of the original paper, that employing stronger augmentation on majority classes and milder augmentation on minority classes yields superior model performance compared to the opposite strategy."*
> #### *"Our analysis confirmed both inferences, indicating that CUDA enhances feature representation by promoting balance in inter-class distances and diminishing the intrusion of foreign instances by reducing intra-class distances.".*

---

> ### Author Response · Authors · 2024-04-05
> **Response to reviewer MBg6 part 3/3**
>
> #### **5) Change regarding absence of tags on red and blue curves in Figure 5**
> #### We have added the tags for red-vanilla and blue-cuda in the image itself. We have also stated the observation we get from figure 5 in its caption. We added detailed citations to all figures in the paper to let readers understand the experiment and the result solely based on the figure and the citation.
>
> #### **6) Change regarding over claim for section 4.1.1**
> #### We have changed the claim to *"a consistent increase in accuracy across all 5 models for the imbalance ratios 100 and 50 when paired with CUDA compared to the Vanilla edition. However for the imbalance ratio 10 we see a minor improvement in accuracy for all models except when CUDA is paired with CE+CMO."*

---

### Review · Reviewer_89HM · 2024-03-17

**Summary Of Contributions:**

This is a reproducibility paper for "CUDA: Curriculum of Data Augmentation for Long-Tailed Recognition" (Ahn et al., 2023). The paper reproduces aspects of the original paper as well as probes new dimensions of the approach.

**Audience:**

Yes

**Claims And Evidence:**

Yes

**Requested Changes:**

Required:
* Fix clarity issues noted in weaknesses. Please fix the writing to make the paper more clear. Update the figures so they look clean and have summarizing captions.
* Please state what the intention/goal of each section is so the reader can understand what the takeaway is. The reader should be able to tell if the results are positive or negative without having to reference the original paper.

Strengthen:
* Quantify the degree of reproducibility achieved

**Strengths And Weaknesses:**

Strengths:
* The paper reproduces what seems to be a substantial portion of the paper and probes additional points in the design space.
* Experiments and Code are released for reproducibility.

Weaknesses:
* It is difficult to understand where the paper is going or what the conclusions are. The figure captions don't help with understanding the figures. It is difficult to understand what is expected behavior in the reproduction, what is novel behavior, when deviations from the paper occur, etc.
* Writing and presentation style can be improved. Paper has many typos and Latex errors that significantly impact clarity of the paper. Commas and periods tend to have unnecessary space around them, while places that require a space do not have one. Informal words inappropriate for technical writing such as "gonna" used. Capitalization is used inappropriately.
* Figures are messy and have inconsistent style. Figure 2 is, for example, a photo that seems directly taken from the CUDA paper. Figure 3 is an image of a table rather than a latex table. Figure 4 requires zooming to see details. Figure 5 has an illegible x-axis. Other figures are small, have visible borders, and in general can be improved with how information is presented. Figure 7, for example, uses 2 different colors (red/green) for the exact same concept, CUDA.
* First 4 pages are essentially a summary of the CUDA paper, but the reader will likely have to read that paper anyway to make sense of the reproduction. Section 3.1.2 seems less clear than the original explanation, partially because there appear to be typos and the presentation in bullet form lacks motivating explanation.
* Code could be organized further.

---

> ### Author Response · Authors · 2024-04-09
> **Response to reviewer 89HM part 1/3**
>
> #### We are very grateful for your constructive comments. We express your concerns as we understood and answer them as follows. We were able to improve the quality of the paper based on your review with the intention of making the paper more clear to the audience. We answer the questions by reordering them in the order in which we thought they were highly correlated to help your understanding. Please consider re-evaluating the paper if you are satisfied with the rebuttal. Please let us know through further comments if there are any misunderstandings or if you have any additional questions or any additional changes to be made in the paper.
>
> #### **1)Concern regarding Lack of Clarity in section 3.1.2**
> #### Section 3.1.2 has been restructured to make it more clear. The typos in Latex have been removed to the best of our knowledge. The motivation towards presenting the data on a bullet form was to explain the algorithm of CUDA and to introduce the 3 hyperparameters on a step by step basis. We used the example of $L^2_1$ the LOL score for a specific class 1 after the second iteration to do so. We track how $L^3_1$ the LOL score for a specific class 1 after the third iteration is found out from $L^2_1$ and in the process introduce the hyperparameters T-number of test samples(sampling) and $\\gamma$-acceptance rate(adjustment). We further explain how $\\rho$-augmentation probability is used to find the updated dataset $D_{cuda}$.
>
> #### In each such step we added a sub-title for example *Calculation of $V_{correct}$* , we corrected latex errors for example *$1_{{f_\theta(\mathcal{O}(x ; l))=c}}$* had a typo because we earlier used a  special character inside the indicator function which has now been replaced with *1*,we gave a generalized description of what happens in that step and stated what happens for our specific example of  $L^2_1$ i.e (c=1 in this case).
> -  #### *Calculation of $V_{correct}$: This involves summing the indicator function $1_{{f_\theta(\mathcal{O}(x ; l))=c}}$ for all samples in $D^{'}c$, where $\mathcal{O}(x ; l)$ represents the application of augmentation strength $l$ to sample $x$. If the model correctly predicts class c (c=1 in this case), the function evaluates to 1. In essence, $V_{correct}$ quantifies the number of correct predictions among the $T(l+1)$ samples in $D^{'}_c$.*
>
> ___
>
> #### **2)Concern regarding improper figures**
> #### We made the following changes to the figures based on your comments
> - #### Each subfigure in Figure 4: Evolution of LOL scores now occupies 0.33 of the page width, an increase from 0.19.We couldn't expand the size any further to ensure that the primary material of the paper didn't exceed 12 pages.
> - #### The x axis of the subfigures in Figure 5: Variation of weight L1 norm has been modified to enhance its legibility. We really regret not noticing the unclear x-axis sooner.
> - #### In figure 8 (formerly figure7) each of the sub-figures have green columns to represent the complete version of CUDA. Other colors are used to denote CUDA versions that do not have a specific component or employ other augmentation techniques. We confined the y axis to a smaller range (for example, 38-50 instead of 0-60) to make the difference in performance between the versions more obvious.
> - #### Sub-figures having visible borders across all figures have been modified to the best of our knowledge.
>
> ___
>
> #### **3)Concern regarding lack of comprehensive citations**
> #### Citation of each figure have been updated. Each citation contains a title for the figure/sub figure, a brief description for each of the axis and the summarizing observation we get from that figure.
>
> #### *Figure 4: Evolution of LOL score in the order CE,CE-DRW,LDAM-DRW,BS,RIDE over the epochs.The x-axis represents the epochs, with the 200th epoch positioned on the rightmost side of each graph. The y-axis displays the classes, arranged in descending order based on the number of samples in each class.The intensity of the color of the heatmap represents the strength of augmentation.It is evident that during training the majority classes(0-49) have a stronger augmentation compared to the minority classes(50-99).*

---

> ### Author Response · Authors · 2024-04-09
> **Response to reviewer 89HM part 2/3**
>
> #### **4) Concern regarding not stating intention of each section**
> #### We have modified each subsubsection in subsection 4.1 and 4.2 to clearly state each of the following wherever applicable
> - #### A brief summary of the metric being measured or the general reason for conducting the study, as stated in the original paper.
>
> #### *The variance of classifier weight norm is usually used to measure how balanced the classifier is considering the input from a class-wise perspective(Kang et al., 2021).A lower variance in weight L1 norm indicates that the classifier assigns similar importance to all classes.*
>
> - #### The various scenarios in which we do the experiment.
>
> #### *We plot the progression of LOL scores (the strength of augmentation) for various classes across five models: CE, CE-DRW, BS, LDAM-DRW, and RIDE*
>
> - #### The anticipated outcome from the original research, the result of our replication of that study, and the figure associated with that study.
>
> #### *The anticipated result was a consistent reduction in the value of variance of weight L1 norm on using CUDA across all 4 models. In our reproduction of this study we observe that there is a significant decrease in the standard deviation of weight L1 norm when we use CUDA compared to the vanilla version from Figure 5 validating the results of the original study*
>
>
> - #### Any assertion we validated or inference we drew from reproducing the experiment.
>
> #### *The heat maps presented in Figure 4 validate the assertion that "Stronger augmentation on majority classes and weaker augmentation on minority classes yields better performance"*
>
> - #### In section 4.1.1, where we evaluate the validation accuracy of CUDA across various combinations of models and imbalance ratios (which was one of the three motivations for our paper), we report the largest deviation (2.4%) we observed when compared to the original paper. Because the original study does not claim any quantifiable results, we did not measure deviation in any subsequent experiments.

---

> ### Author Response · Authors · 2024-04-09
> **Response to reviewer 89HM part 3/3**
>
> #### **5)Concern regarding quality of technical writing**
> #### We utilized grammar correction tools to identify and rectify any grammatical errors, and manually revised such sentences to enhance the clarity of the paper. Additionally, we addressed LaTeX errors highlighted by Overleaf and ensured that all formulas were cross-checked with the original paper to the best of our ability. Furthermore, we refined the tone of the paper to align it more closely with academic standards and minimize informality. We have changed the spacing for commas and periods to (end. Begin) or (end, begin) across the paper. However, should any issues persist in the quality of the writing, we would greatly appreciate it if you could specify them in further comments.
>
> #### **6)Concern regarding figure 2 and 3**
> #### Figure 2 and Figure 3 present specific challenges. Figure 2, depicting the CUDA algorithm, was directly extracted as a photograph from the original paper, as the TMLR style file lacks a format resembling the algorithm layout in the original paper. Similarly, Figure 3, illustrating a comparison of validation accuracy, appears as an image of a table rather than a LaTeX-generated table, as the TMLR style file does not include features for highlighting table rows. While we could have overcome these challenges by utilizing an additional style file, we were concerned that such an action might contravene established rules. Moreover, this could potentially push the main body of the paper beyond the 12-page limit(While we have some control over the size of images, we lack the same level of control over the size of the text).We would be grateful to know your opinion on this regard.
>
> #### **7)Concern regarding Quantifying degree of reproducibility**
> #### In section 4.1.1, where we evaluate the validation accuracy of CUDA across various combinations of models and imbalance ratios (which was one of the three motivations for our paper), we report the largest deviation (2.4%) we observed when compared to the original paper. Because the original study does not claim any quantifiable results, we did not measure deviation in any subsequent experiments.
>
> #### **8)Concern regarding organizing code**
> #### The repository has been restructured to make it easier for readers to find corresponding readings, graphs, or code for each subsection in the results section.
>
> #### **9) Request to read "Response to reviewer MBg6 part 2/3"**
> #### In our response to another reviewer, MBg6, we emphasized the modifications undertaken to enhance the clarity of the paper's motivation for readers. We would be really grateful if you could read the rebuttal.

---

### Review · Reviewer_jwo3 · 2024-04-04

**Summary Of Contributions:**

This work introduce CUDA, which aims at improving the accuracy of Long-Tailed Recognition (LTR) models, particularly with imbalanced datasets. CUDA introduces a class-wise data augmentation technique that employs 'Strength based augmentation' and a score for 'Level of Learning (LOL)' to generate augmented samples. The strength-based augmentation strengthens the augmentation for majority classes and weakens the augmentation for minority classes. The LOL score adjusts dynamically based on the model's prediction success, leading to a form of curriculum learning strategy.

Experiments across different LTR models have shown model validation accuracy improvements when CUDA is used. Moreover, the study provides analytical insights and conducts various evaluations on the approach's effectiveness across different imbalance ratios, hyperparameters, and comparative performance with other augmentation methods. However, the experiments also show that CUDA would not incorporate performance improvement on datasets with lower imbalance, suggesting that it doesn't yield significant improvements on balanced datasets. Furthermore, the results indicate the significant role of cutout augmentation, even overshadowing that of curriculum learning or class-wise scores on the accuracy gain observed when using CUDA.

**Audience:**

Yes

**Claims And Evidence:**

Yes

**Requested Changes:**

Please address weakness points that I mentioned.

**Strengths And Weaknesses:**

Strength:
S1. This work studies a well-motivated problem (long-tailed recognition). this study introduces an algorithm that uses strength-based augmentation in conjunction with a Level of Learning (LOL) score to generate augmented samples. This approach addresses the distinct needs of majority and minority classes, tailoring the complexity of augmentations to achieve improved performance.

S2. Experiments that evaluate CUDA on various LTR models such as CE, CE DRW, LDAM DRW, BS, and RIDE demonstrate the versatility and broad applicability of CUDA on a wide range of base models. This cross-model testing facilitates a deeper understanding of effectiveness with CUDA.

S3. Additional analysis also explores the contribution of Curriculum Learning and class-wise scores to the model's performance. This in-depth analysis allows identifying key factors that influence the outcome and lead to better fine-tuning of the algorithm.

Weakness:

W1. While providing improvements on imbalanced datasets, the efficacy of CUDA diminishes as it is used with datasets having lower degrees of imbalance. This suggests that the advanced features of CUDA might not be essential or beneficial when dealing with balanced datasets. Authors should consider backup strategy for CUDA when handling the datasets not much imbalanced, or provide a criteria on the class-wise distribution for methodoldy selection.

W2. The role of cutout augmentation was found to be significant, potentially overshadowing the contribution of Curriculum Learning and class-wise scores. Such dependency may limit the generalizability of the CUDA approach across different contexts.

---

> ### Author Response · Authors · 2024-04-04
> **Needing clarification on the weaknesses mentioned**
>
> We are very grateful for your constructive comments. To remove any potential miscommunication\misunderstanding we seek to state once more that our paper is a reproduction study of the paper CUDA: Curriculum of data augmentation for Long-tailed Recognition https://openreview.net/forum?id=RgUPdudkWlN. We didn't "introduce" CUDA rather we had reproduced the experiments stated in the original paper and further analyzed CUDA beyond the original paper. In light of this, both the weaknesses that you stated are the drawbacks in the original paper/implementation which we discovered by analyzing imbalance ratio(section 4.2.2) and cutout(section 4.2.3).We believe just like you have stated that these results demonstrate that CUDA doesn't perform well in all scenarios. We will be modifying the Discussion section to empathize the weaknesses of CUDA more. If there is any query that we misunderstood or additional questions, we look forward to being able to answer them with further comments from you.

---

> > ### Comment · Reviewer_jwo3 · 2024-04-06
> > **Thanks for your clarification.**
> >
> > Thanks for your clarification. I understand that:
> > 1. Authors clarified that their work is a replication study of the CUDA paper, not an introduction.
> > 2. Weaknesses observed were inherent to the original study, highlighting areas for improvement.
> > 3. Authors acknowledge the limitations and plan to emphasize CUDA's weaknesses in the discussion section further.
> >
> > In this case, my further comments are:
> > 1. it looks like an empirical study on replicating CUDA on LTR problems, through experimentation. In terms of study design, have you identified any (external/internal) threats to validity in your study (especially when your major hypothesis was set to discovering the performance drawback of CUDA)?
> > 2. I am wondering whether it is possible to setup some assumptions on LTR, (such as the data distribution before or after augmentations). In this way, authors could establish some analytical results on the conditions that CUDA could improve the overall performance.

---

> > > ### Author Response · Authors · 2024-04-12
> > > **Response to reviewer jwo3**
> > >
> > > #### **1) Our major hypotheses and motivations**
> > > #### Our major hypothesis / motivation was not to discover the performance drawback of CUDA. The 3 primary motivations, as stated in the paper, is
> > >
> > > #### *The motivation behind this reproducibility study is threefold: (1) To validate the assertion made by the original authors that employing class-wise augmentation strength can enhance performance of existing LTR models. (2) To confirm the counter-intuitive observation from the original paper that employing stronger augmentation on majority classes and milder augmentation on minority classes yields superior model performance compared to the opposite strategy. (3) To delve deeper into how CUDA effectively addresses the imbalance problem by leveraging insights from prior research on feature representations in long-tailed datasets, a dimension not explored in the original paper.*
> > >
> > > #### Even though, as stated in our previous response, "Needing clarification on the weaknesses mentioned," that we discovered two drawbacks in CUDA, we did not include "discovering performance drawbacks of CUDA" as a primary motivation because we wanted to prioritize the other three motivations, which are already very intriguing. If you insist, we can include additional experiments on the drawbacks of CUDA in the appendix. We would appreciate it if you could share your thoughts on this in subsequent comments.

---

### Comment · Action_Editor_LKTC · 2024-04-04

Hi reviewer jwo3,

I understand your are busy, but please submit your late review for this submission at your earliest convenience. Thank you!

AE

---

### Decision · Action_Editor_LKTC · 2024-05-11

**Recommendation:** Accept with minor revision

**Comment:**

This is a reproducibility paper for "CUDA: Curriculum of Data Augmentation for Long‐Tailed Recognition". I think its quality is generally acceptable as a reproducibility paper. Two reviewers vote for acceptance and one for rejection. More specifically, a positive reviewer has left the following post-rebuttal comments:

> The paper is a reproducibility work, which may be useful for those who are interested in studying related algorithms. That said, the paper is rather complex to read otherwise as it lacks a traditional story and the graphs have a lot of distracting detail. The authors did put an effort to clean up the paper presentation; however, some minor issues persist---Figure 2 is still an image from another paper and there are still some typos when scanning through ("kaggle" and "linked-in" are lowercase). Overall, I do see value in this study if someone is specifically interested in the related algorithms, though the results are more difficult to generalize beyond that.

I would like to accept the submission. Since this submission has a sole author, I strongly suggest the author finding a real researcher to proofread the final version of the manuscript (not an LLM), given that reviewers pointed out the formal writing (language and formatting) issue as a major issue that significantly impacted clarity.

**Audience:**

Yes

**Claims And Evidence:**

Yes